# Transfer of Gold, Platinum and Non-Ferrous Metals from Matte to Slag by Flotation

**Alexey M. Amdur** [1], **Sergei A. Fedorov** [1,2,*] **and Vera V. Yurak** [1]

[1]  Research Laboratory of Disturbed Lands' and Technogenic Objects' Reclamation, Ural State Mining University, 620144 Yekaterinburg, Russia; engineer-ektb@rambler.ru (A.M.A.); yurakvv@m.ursmu.ru (V.V.Y.)
[2]  Institute of Metallurgy of Ural Branch of RAS, 620016 Yekaterinburg, Russia
*  Correspondence: saf13d@mail.ru

**Abstract:** One of the stages of extracting gold and platinum from sulfide materials and circulating slags is the melting stage in ore-thermal and electric furnaces, where the melt is separated into matte and slag. Gold, platinum, and non-ferrous metals are concentrated in the matte. However, a significant portion of them ends up in the slag, which reduces recovery and increases environmental pollution. The main reason for their transition to slag is the flotation of sulfide droplets by gas bubbles, a significant proportion of which occurs during the decomposition of sulfides. Gold and platinum are associated with matte droplets during flotation. Evaluation of adhesion showed that it is large and comparable to the cohesion of these metals. One of the options to reduce the loss of valuable components is to add fluxes to the slag. The influence of calcite and fluorite on the distribution of gold and platinum over the melting products of copper–nickel sulfide materials (matte and slag) has been experimentally studied based on the above theoretical concepts of droplet flotation. When calcite was added to sulfide ore, there was a significant decrease in the sulfur content in the slag (more than 3 times). This, in turn, led to a decrease in non-ferrous metals in the slag by 2–3 times, with gold from 0.45 to 0.29 g/t and platinum from 2.15 to 2.06 g/t. The addition of fluorite to the mixture of copper–nickel matte and model slag ($CaO/SiO_2/Al_2O_3 = 40/40/20$) significantly reduced the sulfur content and non-ferrous metals by 1.5 times, whereas gold was not found in the slag. The decrease in the number of sulfides in the slag is mainly because the listed additives reduce its viscosity. This leads to acceleration of the coagulation of sulfide drops, which are inevitably carried into the slag during flotation, and increases the rate of their settling to the slag–matte boundary, where they merge with the matte mass.

**Keywords:** gold; platinum; sulfide droplet; gas bubble; flotation; flux; calcite; fluorite; losses

## 1. Introduction

The extraction of gold and platinum from sulfide materials is a multi-stage process, including the melting of the prepared concentrate in ore-thermal furnaces. There is when the melt is separated into sulfide and oxide components (matte and slag). Gold, platinum, and non-ferrous metals are concentrated in matte, but a significant part of them ends up in the slag, which leads to metal losses [1,2]. Losses also occur during depletion smelting in electric furnaces of converter slag and slag of suspended smelting furnaces. The distribution of Pt, Pd, Au, and non-ferrous metals between slag and matte, depending on various factors, is considered in [3,4]. It is known that the solubility of Pt and Au in slags is negligible [5]. One of the probable options for the transition of non-ferrous and noble metals into slag is, as our studies show, the flotation of matte droplets containing them with gas bubbles. Most of these bubbles appear as a result of the decomposition of sulfides. In the process of flotation, the bubble must overcome the matte–slag interface and hold the matte drop while floating in the slag. Matte droplets coagulate in the slag. As a result, large droplets will settle to the surface of the slag–matte. To increase the rate

of coagulation and sedimentation of droplets, it is necessary to lower the viscosity of the slag. This is possible by adding fluxes. The flotation of matte droplets in slags of various compositions depend on the partial pressure of oxygen; interfacial tensions in this system were considered in [6–9] and the breakthrough of a gas bubble of the surface of two liquids was analyzed in [10]. The listed questions of the description of flotation as well as the influence of fluxes on the loss of valuable metals have been little studied, but are relevant.

The purpose of the present work is to search for ways to increase the degree of extraction of platinum, gold, and non-ferrous metals, and experimentally study the effect of fluxes–calcite and fluorite on their distribution between matte and slag, estimated by the total metal content in slag and matte, based on theoretical concepts of droplet flotation.

## 2. Materials and Methods

Materials for research are represented by copper–nickel sulfide ore, gold-bearing pyrite ore, a mixture of copper–nickel matte, and model slag ($CaO/SiO_2/Al_2O_3 = 40/40/20$) in a ratio of 5:3. The chemical compositions of the materials are given in Table 1, and the mineral compositions are shown in Table 2. The mineral compositions of the materials were determined to predict their phase composition with a change in temperature in the HSC Chemistry 9.0 software package (Metso Outotec, Helsinki, Finland) [11].

**Table 1.** Chemical composition of the investigated materials.

| Component | Content, wt.% | | |
|---|---|---|---|
| | Copper–Nickel Sulfide Ore | Gold-Bearing Pyrite Ore | Copper–Nickel Matte |
| $SiO_2$ | 5.60 | 10.2 | - |
| $Al_2O_3$ | 2.40 | 0.37 | - |
| CaO | 0.30 | 1.12 | - |
| MgO | 0.40 | 1.32 | - |
| Fe | 54.6 | 37.7 | 57.9 |
| MnO | 0.20 | - | - |
| $K_2O$ | 0.30 | 0.24 | - |
| $Na_2O$ | 0.30 | 0.80 | 0.38 |
| S | 31.3 | 45.1 | 29.8 |
| Cu | 4.67 | 1.83 | 7.02 |
| Ni | 4.28 | - | 4.98 |
| Co | 0.30 | - | 0.15 |
| Au | $0.4 \times 10^{-4}$ | $1.2 \times 10^{-4}$ | $13.1 \times 10^{-4}$ |
| Pt | $1.1 \times 10^{-4} \div 3.1 \times 10^{-4}$ | - | - |

**Table 2.** Mineral composition of the investigated materials.

| Mineral. | Content, wt.% | | |
|---|---|---|---|
| | Copper–Nickel Sulfide Ore | Gold-Bearing Pyrite Ore | Copper–Nickel Matte |
| Qurtz $SiO_2$ | 8.0 | - | - |
| Clinochlor $Mg_6Si_4O_{10}(OH)_8$ | 3.0 | - | - |
| Sericite $KAl_2(AlSi_3O_{10})(OH)_2$ | 2.0 | - | - |
| Olivine $(Fe, Mg)_2SiO_4$ | - | 0.7 | - |
| Enstatite $Mg_2Si_2O_6$ | - | 0.7 | - |
| Dolomite $CaMgCO_3$ | - | 1.3 | - |
| Plagioclase $(Na, Ca)AlSi_3O_8$ | - | 3.3 | - |
| Magnetite $Fe^{2+}Fe^{3+}{}_2O_4$ | - | 12.0 | 4.9 |
| Pyrite $FeS_2$ | 79.6 | - | - |
| Troilite FeS | - | - | 72.4 |
| Pyrrhotite $Fe_7S_8$ | - | 42.8 | - |
| Chalcopyrite $CuFeS_2$ | 5.1 | 20.0 | - |
| Pentlandite $Ni_{4.5}Fe_{4.5}S_8$ | - | 15.0 | - |
| Sphalerite ZnS | 2.0 | - | - |
| Bornite $Cu_5FeS_4$ | 0.1 | - | 10.1 |
| Tetratenite FeNi | - | - | 7.0 |

Gold-bearing pyrite ore is a massive fine-and medium-grained aggregate with a low content of rock-forming (quartz and others) and copper minerals (chalcopyrite). Gold consists of micro-dispersed spherical particles. Copper–nickel sulfide ore is a massive medium and coarse-grained aggregate with an uneven content of copper and nickel minerals (chalcopyrite and pentlandite). The platinum in it is represented by sperrylite ($PtAs_2$), the size of which rarely exceeds 10 μm. Copper–nickel matte was obtained as follows. Copper–nickel sulfide ore was melted in corundum crucibles (volume 100–150 mL) in a resistance furnace with a graphite heater at a temperature of 1300 °C and holding for 30 min. Heating to the specified temperature lasted 1.5–2 h, and cooling took place together with the furnace for 1–2 h. After that, the matte was separated from the slag, the matte was crushed to a particle size of less than 0.071 mm, and re-melting was performed (to completely remove the slag particles). The re-melting conditions are like the first. The model slag was obtained by mixing chemically pure powders of CaO, $Al_2O_3$, and $SiO_2$ (grain size less than 0.071 mm) in the above ratio. The resulting charge was melted in a corundum crucible in a resistance furnace with a graphite heater at a temperature of 1350 °C and holding the melt for 20–30 min. The slag was removed from the crucible and ground to a grain size of less than 0.071 mm.

The samples were preliminarily ground to a powder state. Dispersed matte droplets in the slag were not separated from it. The mineral composition was determined by X-ray phase analysis on an XRD 7000C X-ray diffractometer (Shimadzu Corporation, Kyoto, Japan). The fraction of each phase was determined using the Crystal Impact Match program [12]. The chemical composition of the materials was found using the Spectroflame Modula S (Spectro Analytical Instruments GmbH, Kleve, Germany) inductively coupled plasma atomic emission spectrometer. This spectrometer allows the content to be determined with an accuracy of 0.1 ppm. The detection limit for chemical elements is 0.1–0.3 ppm (g/t).

After melting, the material was removed from the crucible. The resulting ingots of matte and slag were separated. Several heats were carried out. Part of the material, both matte and slag, was abraded to a powder state, and the above analyzes were carried out. From the other part, polished sections (3 × 2 cm in size) were made, which were then ex-amined under an Axio Image optical microscope (Carl Zeiss Microscopy GmbH, Jena, Germany) and a Tescan Vega 3 scanning electron microscope (Tescan Orsay Holding, a.s., Brno-Kohoutovice, Czech Republic) equipped with an Oxford Instruments X-act energy dispersive analyser (Oxford Instruments, Abingdon, UK). SEM photography was performed at a high vacuum, with a voltage of 20 kV and in a backscattered electrons mode (contrast mode by average atomic number). The energy dispersive analyzer allows the determination of chemical elements with an accuracy and detection limit of 0.1 wt.%. Preliminarily, polished sections were placed on a metal holder and deposited with a thin layer of graphite under vacuum. Polishing was carried out to a mirror surface. Polished sections were made to study the chemical and phase composition, in particular, droplets and particles containing gold and platinum. To facilitate the search on microscopes for phases containing gold and platinum, in a number of cases, Au and Pt were added to the studied ore, i.e., Pt in the form of specially prepared spongy platinum with a particle size not exceeding 100 μm, in an amount from 1.0 to 2.5 kg/t (in copper–nickel sulfide ore), and Au in the form of a microdispersed powder (1.3 kg/t, in gold-bearing pyrite ore). The added gold and platinum (during heating, holding at high temperatures, and cooling) interacts with the components of the ore in a similar way to natural ones.

The materials were heated and melted in corundum crucibles in a resistance furnace with a graphite heater in air at a temperature of 1300 °C. Holding time is 30 min, and cooling with the oven is 1–2 h. The heating time of the furnace to the specified temperature is 1.5–2 h. Corundum crucibles are glasses with a volume of 100–150 mL. The melting point of the charge was determined as follows. A W–Re thermocouple in a corundum sheath was in contact with the charge. At the indicated temperature, a break was observed on

the temperature- heating curve, which we interpreted as melting of the charge. This is confirmed by visual observations.

## 3. Results and Discussion

### 3.1. The Form of Finding of Gold and Platinum in the Melting Products of Sulfide Materials

The experiments with a gold-containing pyrite ore showed that, at 1300 °C, all the material melted and separated into matte and slag. Most of the gold is concentrated in matte (content about 1.5 kg/t). After cooling, gold particles are located mainly along the cracks of the sulfide matte minerals (troilite and bornite). The shape of the Au particles is drop-shaped. All of them contain impurities: Cu (12.4–15.8 wt.%), Ag (1.1–6.1 wt.%), and Sb (1.7–5.5 wt.%). The average gold content is 81.0 wt.%. The Au particle size ranges from 2 to 43 μm. Some large gold-bearing particles are heterogeneous in composition and contain at least two more phases. The first is represented by an intermetallic compound (As, Fe, Pt, Sb). It forms wedge-shaped crystals with rhombic sections (Figure 1), up to 4 μm in size. The platinum content in the intermetallic compound reaches 10.0 wt.%. The second phase is an alloy (Sb, Au) with a gold content of up to 40.0 wt.% and platinum (up to 6.0 wt.%). It has an irregular shape and a maximum size of 2 μm. Earlier, we found [13] that the amount of metallic impurities in dispersed particles of ore gold (in particular, in gold-bearing copper-pyrite ore, Table 1) increases with decreasing their size (with a particle size less than 3 μm). In our opinion, the size effect is associated with the amorphization of particles smaller than 3 μm, a sharp change in the structure and properties in comparison with the bulk material and, as a consequence, an increase in the solubility of metals in gold due to an increase in the Gibbs energy. Thus, the increase in the amount of impurities is associated with the size factor, which will take place in both ores and matte.

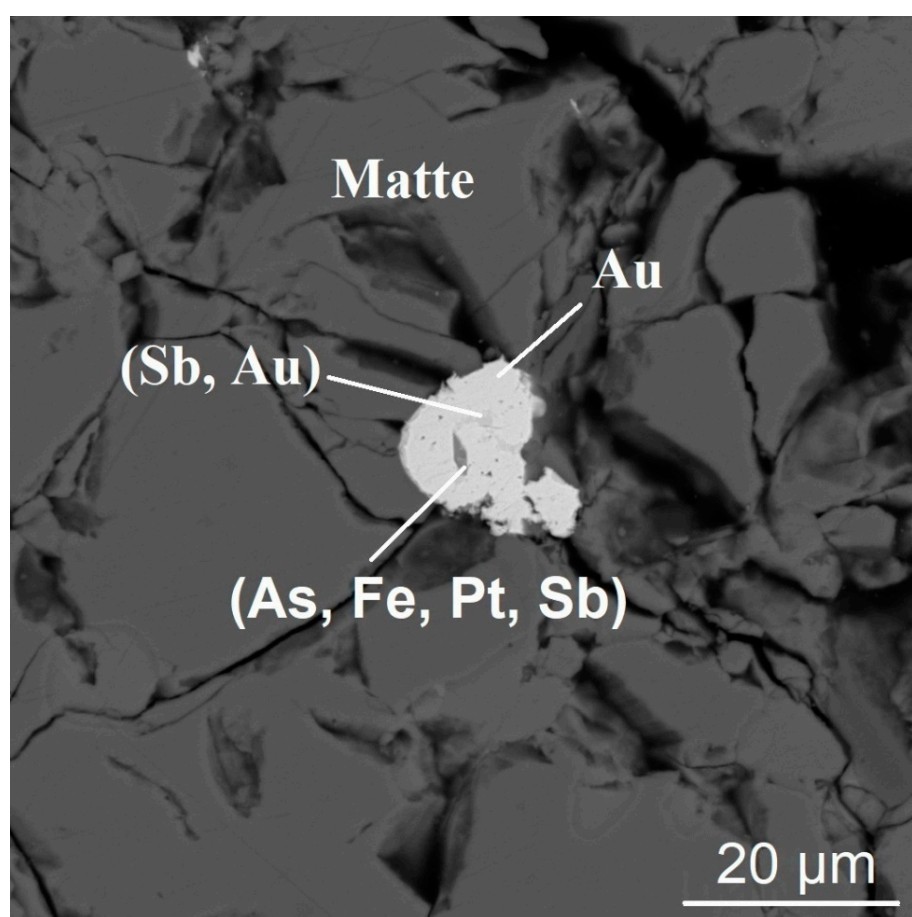

**Figure 1.** Gold-containing particle in matte after melting of gold-bearing pyrite ore at a temperature of 1300 °C.

The slag after melting of gold-bearing copper pyrite ore is represented by a porous greenish-brown glassy mass. It contains many sulfide droplets ranging in size from 2 to 800 microns. The chemical composition of the drops is like that of matte. They are associated with drop-shaped gold particles ranging in size from 1.4 to 17.1 μm, which is smaller than gold droplets in matte, as seen in Figure 2. The gold droplets in the slag have a higher Au content (89.0 vs. 81.0 wt.% in the gold droplets in the matte) and include the same metallic impurities as the gold in the matte. The concentration of total gold in the slag reaches a maximum of 200 g/t.

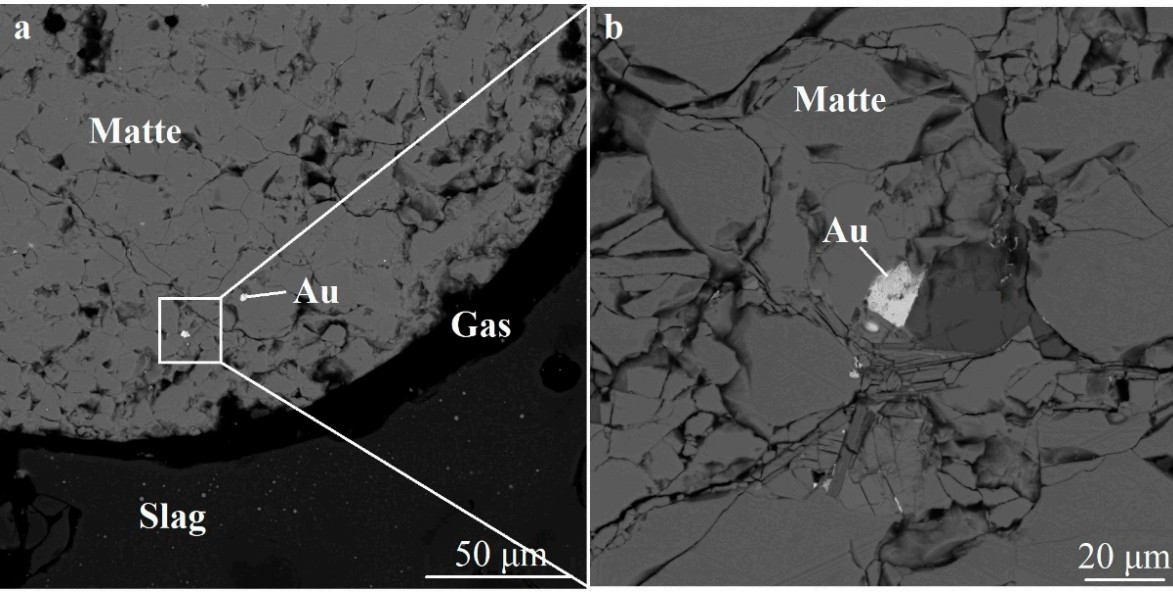

**Figure 2.** Gold-containing particles in sulfide droplets (Matte) located in the slag and attached to a gas bubble (Gas): (**a**) general view of a sulfide droplet (Matte) in slag, (**b**) enlarged fragment of a sulfide droplet with a gold particle. The slag obtained after melting gold-bearing pyrite ore at a temperature of 1300 °C.

Platinum in matte after cooling from 1300 °C of the copper–nickel sulfide ore is in the form of intermetallic compounds and, probably, chemical compounds with Fe and Ni. This intermetallic compound are acicular formations with a length of 20 to 500 microns and a thickness of up to 10 microns (at 1300 °C), Figure 3. These formations are intergrowths of prismatic crystals, which, as shown by chemical analysis using an energy-dispersive attachment, are PtFe intermetallic compounds (content Pt is 69–72 wt.%). The crystals are in a double metal shell. After cooling from 1200 °C, the PtFe intermetallic crystals have a skeletal shape, smaller size (thickness on average 2–3 μm), and contain less Pt (from 47 to 63 wt.%) compared to the samples after cooling from 1300 °C.

The needle formations of the PtFe intermetallic compound contain Cu and Ni impurities. They are typical for this alloy [14]. The platinum content decreases, as in the case of microdispersed Au particles, with a decrease in the thickness of the PtFe formations.

The study of oxide melt (slag) during melting of copper–nickel sulfide ore containing platinum showed the following. The slag is porous due to the release of gas bubbles containing sulfur. These bubbles, as can be seen from the micrographs, float matte droplets up to 1.5 mm in diameter. Irregularly shaped intermetallic particles containing Pt, Fe, and Ni, no more than 5–7 microns in size and found in the surface layers of the slag. They are associated with these drops, as seen in Figure 4.

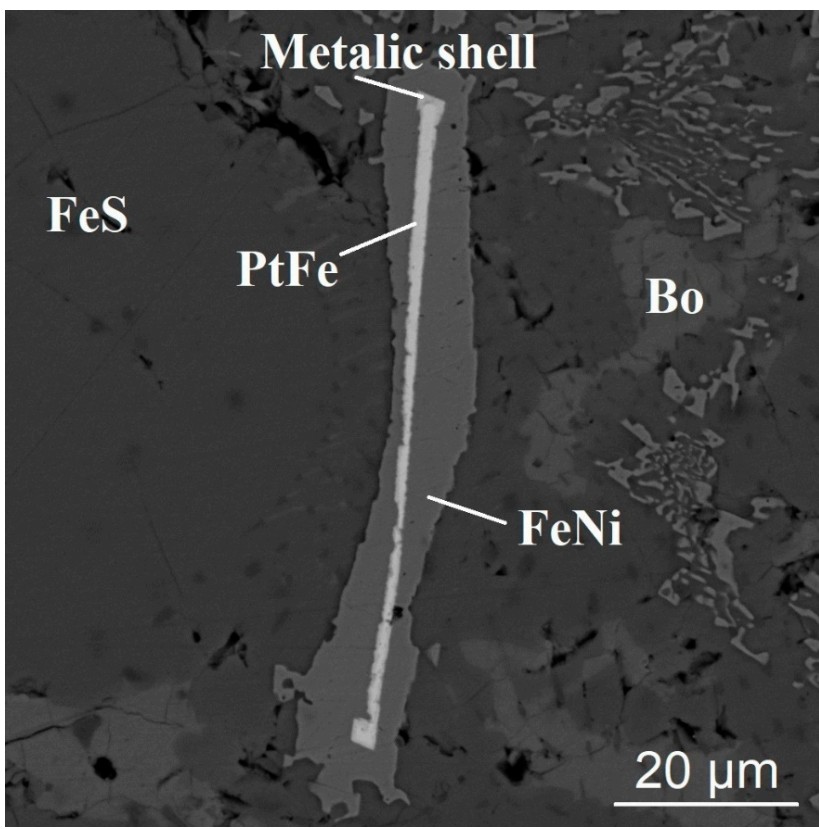

**Figure 3.** Needle formations of the PtFe intermetallic compound in a double shell in a copper–nickel matte. Matte is obtained after melting a copper–nickel sulfide ore at a temperature of 1300 °C.

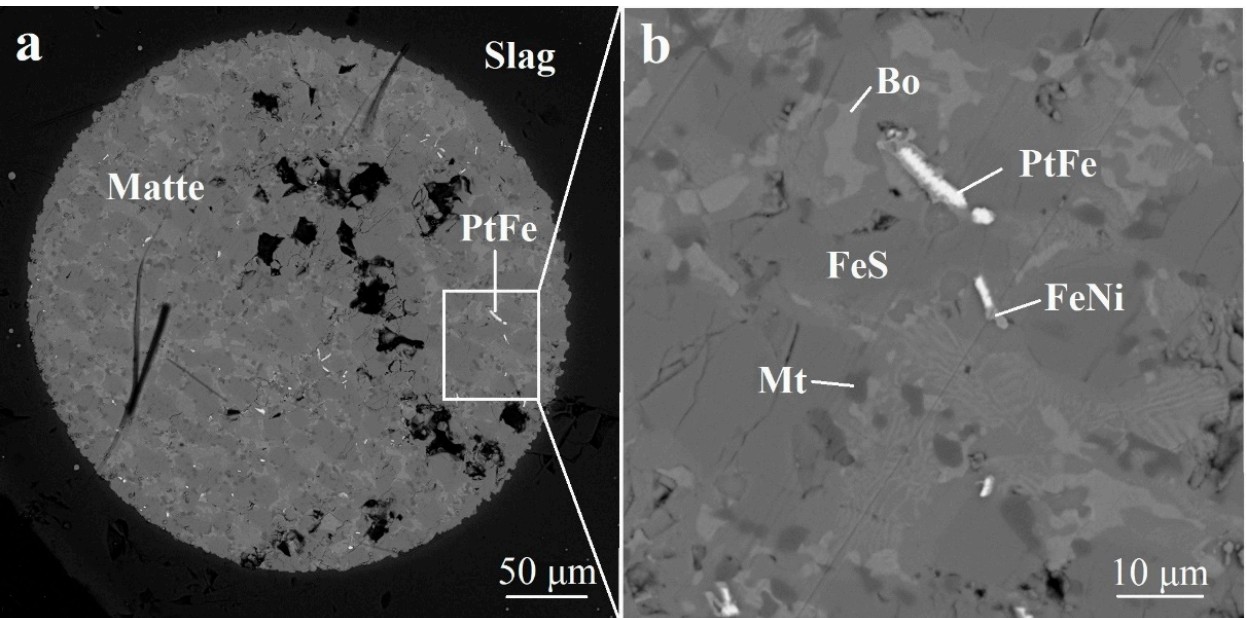

**Figure 4.** Needle-like PtFe particles in a matte drop that is in the slag: (**a**) general view of a matte drop in the slag, (**b**) enlarged fragment of a drop with platinum-containing particles. The slag obtained after melting copper–nickel sulfide ore at a temperature of 1300 °C.

### 3.2. Flotation of Matte Droplets in an Oxide Melt

The main reason for losses is the flotation of matte droplets containing gold and platinum by gas bubbles. They appear as a result of the decomposition of sulfides during the melting of sulfide materials. From the point of view of physical chemistry, flotation means the floating of bodies of higher density than a liquid on its surface under the action of interfacial tension forces. In the process of flotation, the bubble must overcome the matte–slag interface and hold the matte drop while floating up in the slag. For the bubble to float up with a drop of matte in the slag, two conditions must be met.

1. The droplets are held on the bubbles by interfacial tension, which can be called coupling force. It is equal to $2\pi r \sigma_{m\text{-}s} \sin\theta$ ($\theta$ is the angle between the tensions at the matte–slag $\sigma_{m\text{-}s}$ and matte–gas $\sigma_m$ interface) and $\sigma_{m\text{-}s}\sin\theta$ is the vertical component of the interfacial tension at the matte–slag interface. To prevent the drop from breaking away from the bubble, the coupling force of the matte drop with the bubble must be greater than its gravity $P$. The first condition is expressed by the inequality:

$$2\pi r \sigma_{m\text{-}s} \sin\theta > c\frac{4}{3}\pi r^3 \rho_m g \tag{1}$$

where $r$—radius of matte drop, $\rho_m$—matte density, and $c$—fraction of the droplet volume from the full sphere.

The maximum radius of matte droplets that can float by bubbles, depending on the properties of the media and the characteristics of the matte–slag interphase boundaries, according to calculations by inequality (1), is 1.8–4.2 mm. Upon reaching the slag–gas boundary, droplets with a radius of less than 3 mm can be held at this boundary by the forces of the surface tension of the slag.

2. The buoyancy force of Archimedes applied to the bubble ($F_a$) must be greater than the gravity of the drop $P$. Otherwise, the matte drop will separate from the bubble. This condition determines the ratio of the matte drop radius $r$ to the radius of the gas bubble $R$ carrying it:

$$F_a \geq P \tag{2}$$

Or

$$\frac{4}{3}\pi R^3 (\rho_s - \rho_g) g \geq c\frac{4}{3}\pi r^3 (\rho_m - \rho_s) g. \tag{3}$$

where $\rho_s$—slag density, and $R$—bubble radius.

From condition (3), we obtain the following ratio of the radii of the droplet and bubble:

$$\frac{r^3}{R^3} \leq \frac{\rho_s - \rho_g}{c(\rho_m - \rho_s)} \leq \frac{2700}{c1700}, R \geq 0.7 \div 0.9r \tag{4}$$

As the droplet size $r$ decreases, the coupling force decreases in proportion to $r$, and the gravity is proportional to $r^3$. The prevalence of surface forces over gravity is characteristic of all dispersed systems [15]. Therefore, micron matte droplets will almost always float. In the slag, there is a collision of matte droplets and, as a result, coagulation. For example, a bubble with a volume of $V_g$ is capable of lifting gold droplets whose volume $V_{Au}$ does not exceed $1/19 \times V_g$. In our experiments, the gold content was about $4 \times 10^{-6}$ g/g of the melt; the radius of the initial particles was 3 µm; the radius of the bubbles reached 1 mm; and the thickness of the melt was 5 cm. In this case, the bubble on its way upward must collect and bring to the surface of the melt about 44 thousand gold droplets. Then, they are combined by coagulation. As a result, large droplets are formed (up to 300 microns in diameter). According to the Stokes formula, the bubble float speed in the oxide melt is high and amounts to about $6.7 \times 10^{-3}$ m/s. Thus, the flotation of dispersed droplets in melts proceeds at high speeds and leads to their significant enlargement, in contrast to the flotation of solid particles. In the latter case, only the formation of fragile associations is possible. The rate of coagulation of drops, by analogy with colloidal systems, is proportional to temperature and inversely proportional to the viscosity of the slag and

the concentration of drops. The large droplets formed will settle in the slag and end up in the matte again. The deposition rate is inversely proportional to the viscosity. For drops with a radius of 5 μm, depending on the viscosity of the slag, it is small and varies within the range of $0.1 \div 1.2$ μm/s. Therefore, they will remain in the slag. A schematic of the process at the slag–matte interface is shown in Figure 5. Thus, in order to reduce the loss of valuable components, it is necessary to increase the rates of coagulation and precipitation of matte drops in the slag, reducing its viscosity. This is possible by adding fluxes.

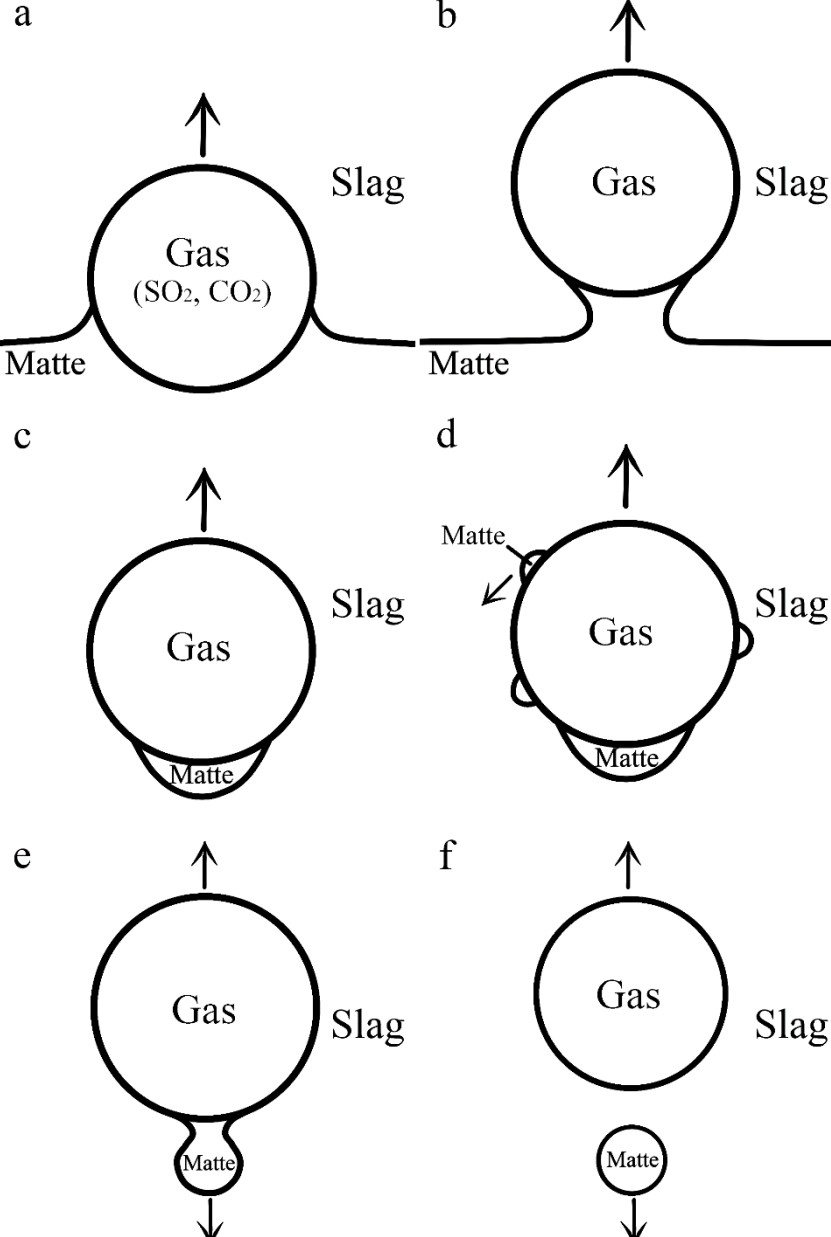

**Figure 5.** Stages of passage of a gas bubble through the matte–slag interface: (**a**) exit of a gas bubble to the matte–slag interface; (**b**) transition of a gas bubble through the matte–slag interface; (**c**) floating up of a gas bubble with a sulfide drop; (**d**) collection by the gas bubble of other dispersed matte droplets encountered on its way; (**e**) separation of the enlarged sulfide droplet from the gas bubble; (**f**) deposition of a sulfide droplet and floating of a gas bubble.

Since drops of gold and solid platinum particles are carried into the slag along with the matte drops, their adhesion $W_A$ to matte was evaluated. Both metals interact with the matte components. Therefore, we are talking about dynamic contact angles and dynamic

(nonequilibrium) adhesion. As for the drops of gold, it does not matter whether the dynamic or equilibrium values of the tensions act on it, as only the magnitude of these tensions is important. The adhesion for gold–matte–slag and platinum–matte–slag systems was calculated using the Dupre equation:

$$W_A = \sigma_{Me\text{-}m}(1 + \cos\theta) \tag{5}$$

where $\sigma_{Me\text{-}m}$—metal–matte interfacial tension.

The values (Table 3) included in Equation (5) were found from photographs of the gold–matte–slag and platinum–matte–slag boundaries. To facilitate the search on microscopes containing platinum and gold phases, Pt and Au, as already indicated, were added to the investigated materials. The photographs of the boundaries were used to determine the angles $\theta$ between the metal–slag and metal–matte interfacial tensions, which are included in the Dupré equation.

**Table 3.** Calculation data.

| $\sigma_{Au}$, N/m | $\sigma_{Pt}$, N/m | $\sigma_{m1}$, N/m | $\sigma_{m2}$, N/m | $\sigma_{s1}$, N/m |
|---|---|---|---|---|
| 1.02 | 1.95 | 0.36 | 0.35 | 0.41 |
| $\sigma_{s2}$, N/m | $\sigma_{m\text{-}s1}$, N/m | $\sigma_{m\text{-}s2}$, N/m | $\theta_{Au}$ | $\theta_{Pt}$ |
| 0.45 | 0.11–0.14 | 0.05–0.11 | 19.5° | 24–37° |

Note: $\sigma_{Au}$ is the surface tension of liquid Au; $\sigma_{Pt}$ is the surface tension of solid Pt; $\sigma_{m1}$ is the surface tension of copper matte; $\sigma_{m2}$ is the surface tension of copper–nickel matte; $\sigma_{s1}$ is the surface tension of copper slag; $\sigma_{s2}$ is the surface tension of copper–nickel slag; $\sigma_{m\text{-}s1}$ is interfacial tension of the system "copper matte–slag"; $\sigma_{m\text{-}s2}$ is interfacial tension of the system "copper–nickel matte–slag", $\theta_{Au}$ is the angle between the interfacial tensions gold–slag and gold–matte; $\theta_{Pt}$ is the angle between the interfacial tensions platinum–slag and platinum–matte.

The work of adhesion is ~1600 mJ/m$^2$ for the gold–matte system and is ~3000 mJ/m$^2$ for platinum–matte. The obtained values are very significant and are comparable with the work of cohesion for gold and platinum. Consequently, these metals are firmly adhered to the matte drops and move along with them in the slag. The adhesion values also agree in order of magnitude with those calculated based on the theory of dielectric formalism in the absence of a gap between the surfaces of metals: Cu/Al—3100, Fe/Cu—4200 mJ/m$^2$. With an increase in the vacuum gap between the surfaces up to 3 A°, the adhesion values for different pairs of metals approach each other and become equal to ~400 mJ/m$^2$ [16].

The fact that gold and platinum end up in the slag together with the associated matte drops is confirmed by the following experimental data. It was found that the amount of gold (Au) in the slag increases with an increase in the sulfur content in it, which is proportional to the matte mass, and reaches a constant value at high S contents, as shown in Figure 6. The output (Au) to a constant value is associated with a small content of gold in the processed materials; not all matte drops passing into the slag are likely to contain it. Since copper, nickel, and cobalt are in the form of sulfides, they also pass into the slag as part of matte droplets by flotation. Indeed, in accordance with Figure 7, their proportion in the slag from the total content, such as Pt and Au, is proportional to the amount of sulfur in the slag. According to the data of chemical analysis, 5 to 35% of the total mass of platinum, 12 to 19% of gold, 7.5% of copper, and 9.5% of nickel pass into the slag. In industrial slags in ore-thermal furnaces when melting on copper–nickel matte [17], the content of platinum ranges from 0.3 to 1 g/t, and gold ranges from 0.1 to 0.4 g/t [1,18,19]. In the slags formed after the processing of copper–nickel sulfide ores, platinum is in the form of alloys with iron and nickel [20]. When lime was added to the initial charge as a flux, the value for copper decreased to 3%, and for nickel it decreased to 3%. This is consistent with production data: 1–3% for Cu and 1–2% for Ni [2,18].

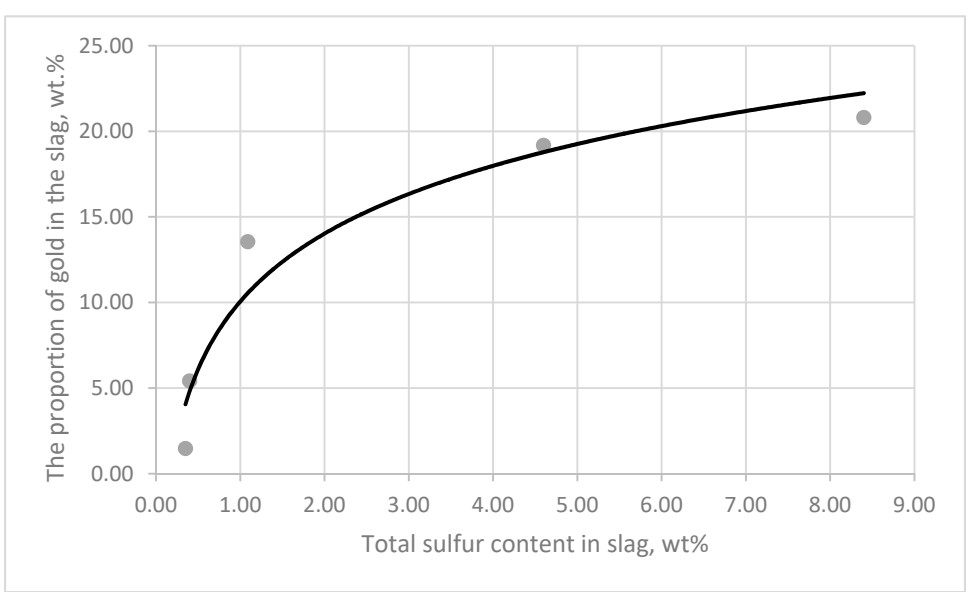

**Figure 6.** The dependence of the proportion of gold transfer from matte to slag on the total sulfur content in the slag.

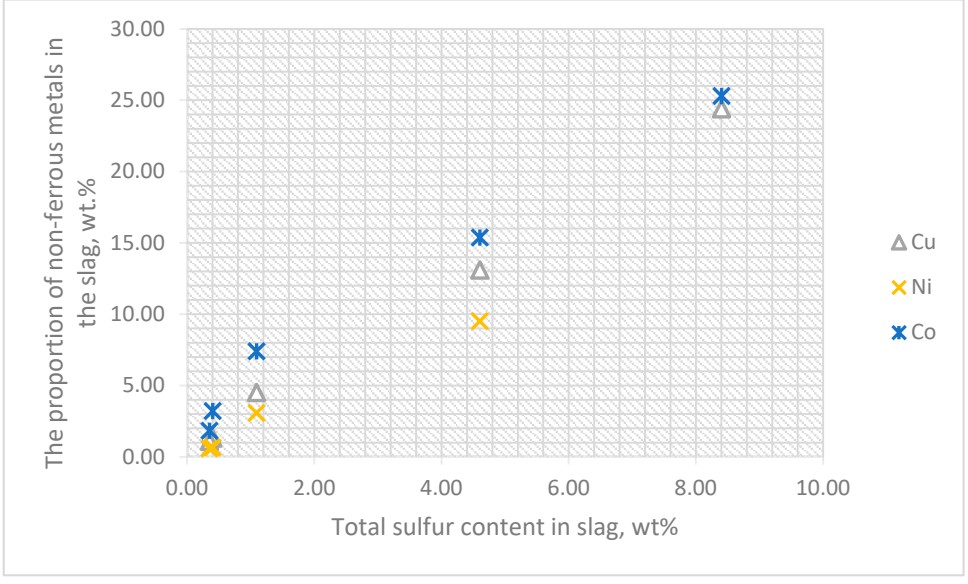

**Figure 7.** The proportion of non-ferrous metals in the slag, depending on the total sulfur content in it.

### 3.3. Influence of CaCO₃ and CaF₂ Fluxes on the Distribution of Gold, Platinum and Non-Ferrous Metals between Matte and Slag

The effect of fluxes, which reduce the viscosity of the slag, on the distribution of gold, platinum, and non-ferrous metals over the melting products of copper–nickel sulfide materials (matte and slag) has been experimentally studied. For the experiments, two fluxes were selected: calcite ($CaCO_3$) and fluorite ($CaF_2$). The first of which was added in an amount of 10 wt.% to the initial copper–nickel sulfide ore, and the second to a mixture of copper–nickel matte and model slag in the amount of 10 wt.%. The choice of fluxes is justified as follows. In the metallurgical production of non-ferrous metals, technologies with the addition of calcite in the form of limestone are known [17]. Calcite releases large amounts of carbon dioxide when heated. Despite the fact that gas bubbles lead to flotation of gold and platinum-containing sulfide droplets and their transition into slag, they also accelerate coagulation and subsequent precipitation of dispersed matte droplets in the slag due to mixing of the melt. CaO formed during the decomposition of calcite lowers the

melting point of the slag and its viscosity. CaO also leads to partial desulfurization of the matte. Slags containing fluorite have low values of surface tension as well as low melting points and viscosity [21].

In order to predict the phase composition, we analyzed possible chemical reactions in sulfide ore upon heating using the Equilibrium Compositions module of the HSC Chemistry 9.0 software package under reducing conditions (with a carbon content of 1.8 wt.%) with varying contents of calcite and fluorite. It was found that gold and platinum during heating in the range 0–1300 °C can be in the following form: metallic forms of Au and Pt; sulfide PtS; and intermetallic compounds, including AuCu, PtFe, $PtFe_3$, and PtNi. Non-ferrous metals are mainly present in the sulfide form. The addition of calcite and fluorite does not make significant changes; all the listed forms of occurrence of gold, platinum, and non-ferrous metals are preserved, and only their contents change.

Matte and slag after holding at a temperature of 1300 °C and cooling were analyzed for S, non-ferrous metals (Cu, Ni, Co), gold, and platinum. The data are shown in Table 4.

**Table 4.** Content of sulfur, non-ferrous, and noble metals in matte and slag after melting the charge without and with the addition of $CaCO_3$.

| Component | Content of Components in wt.% | | | |
|---|---|---|---|---|
| | Without $CaCO_3$ | | $CaCO_3$ in the Amount of 10% | |
| | Matte | Slag | Matte | Slag |
| Cu | 4.66 | 0.38 | 4.68 | 0.15 |
| Ni | 4.28 | 0.45 | 4.39 | 0.14 |
| Co | 0.11 | 0.02 | 0.11 | 0.008 |
| Pt | $4.1 \times 10^{-4}$ | $2.15 \times 10^{-4}$ | $3.8 \times 10^{-4}$ | $2.06 \times 10^{-4}$ |
| Au | $1.77 \times 10^{-4}$ | $0.45 \times 10^{-4}$ | $1.83 \times 10^{-4}$ | $0.29 \times 10^{-4}$ |
| S | 30.5 | 4.60 | 28.7 | 1.13 |

When $CaCO_3$ was added to sulfide ore, it melted at a temperature of 1260–1270 °C, which is 10–20 °C less when melting the ore without adding flux. Table 4 shows that there was the sulfur content decreased more than three times of that in the slag. This, in turn, led to a decrease in non-ferrous metals in the slag by of 2–3 times, gold from 0.45 to 0.29 g/t, and platinum from 2.2 to 2.06 g/t.

The addition of fluorite lowered the melting point to 1210–1220 °C (melting temperature without adding $CaF_2$ is 1270 °C). The content of sulfur and non-ferrous metals has noticeably decreased by 1.2–3 times, while gold is not found in the slag (Table 5). It is below the sensitivity of the used technique. The influence of $CaF_2$ on the amount of platinum in the slag has not been established, probably due to the insufficient accuracy of its determination.

**Table 5.** Content of sulfur, non-ferrous, and noble metals in matte and slag after melting the charge without and with the addition of $CaF_2$.

| Component | Content of Components in wt.% | | | |
|---|---|---|---|---|
| | Without $CaF_2$ | | $CaF_2$ in the Amount of 10% | |
| | Matte | Slag | Matte | Slag |
| Cu | 7.02 | 0.1 | 7.25 | 0.05 |
| Ni | 4.98 | 0.03 | 5.12 | 0.01 |
| Co | 0.15 | 0.005 | 0.16 | 0.002 |
| Au | $13.1 \times 10^{-4}$ | $0.75 \times 10^{-4}$ | $13.5 \times 10^{-4}$ | Not detected |
| S | 29.8 | 0.41 | 30.9 | 0.34 |

Thus, the studies carried out have shown that the addition of calcite and fluorite to the charge increases the extraction of non-ferrous metals, gold, and platinum into the matte.

This is due to the fact that the listed additives reduce the viscosity of the slag. A decrease in viscosity leads to an acceleration of the coagulation of sulfide droplets carried into the slag during flotation and increases the rate of their settling to the slag–matte boundary, where they merge with the matte mass.

## 4. Conclusions

Gold and platinum in the studied products of sulfide ores smelting are in the form of metallic phases containing metallic impurities. The mechanism of their transition into the slag after melting in metallurgical aggregates has been established: matte droplets, together with associated particles of these metals with sizes of no more than 5–7 microns for Pt and 1–17 microns for Au, are carried into the slag by gas bubbles (flotation).

High values of the calculated values of the adhesion work for the gold–matte (~1600 mJ/m$^2$) and platinum–matte (~3000 mJ/m$^2$) systems indicate the adhesion strength of metals with matte drops.

The stages of the passage of a gas bubble through the matte–slag interface are considered. Analytical expressions describing flotation are obtained. The flotation of micron-sized matte droplets in the slag is inevitable and leads to the loss of valuable components. The gas bubble is capable of carrying sulfide droplets with a radius of up to 4.2 mm into the slag.

To reduce losses, it is necessary to increase the rate of coagulation of droplets and their sedimentation in the slag, decreasing its viscosity. This is possible by adding fluxes. It was found that the introduction of $CaCO_3$ and $CaF_2$ (in an amount of 10 wt.% each) into the initial charge during melting of sulfide materials onto matte made it possible to reduce the content of sulfide drops in the slag and, accordingly, non-ferrous metals, gold, and platinum.

**Author Contributions:** Conceptualization, methodology, writing—original draft preparation, supervision, A.M.A.; software, formal analysis, resources, visualization, S.A.F.; investigation, writing—review and editing, validation, V.V.Y. All authors have read and agreed to the published version of the manuscript.

**Funding:** The research was funded by the Ministry of Science and Higher Education of Russia in accordance with the state assignment for Ural State Mining University No. 075-03-2021-303 from 29 December 2020.

**Institutional Review Board Statement:** Not applicable.

**Informed Consent Statement:** Not applicable.

**Data Availability Statement:** Data is contained within the articles, open sources that mentioned in the reference list, and made up by authors during the experiments.

**Conflicts of Interest:** The authors declare no conflict of interest.

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
