# Peer review of "Transfer of Gold, Platinum and Non-Ferrous Metals from Matte to Slag by Flotation"

_metals, doi:10.3390/met11101602_

Round 1
Reviewer 1 Report
Comments and questions to the authors:
- The title of your manuscript is misleading the readers to kinetics and mass transfer between slag and matte; I suggest that you rewrite it in a more focused/shorter form.
- In the Abstract (rows 13-14), you claim that bubbles appear in smelting because of decomposition of sulphides. This is only one form of gas possibly present in the sulphide smelting environments. Please, re-phrase.
- The Introduction Ch deals a little with the (chemical) solubilities of precious and platinum group metals in smelting slags. Such background information and a clear definition of slag losses as mechanical and chemical would be useful when evaluating the claims and observations made in this study. Such data on chemical dissolution are available in several recent publications, e.g.
- Chen M., Avarmaa K., Klemettinen L., O’Brien H., Shi J., Taskinen P., Lindberg D. & Jokilaakso A. (2021): Precious Metal Distributions Between Copper Matte and Slag at High PSO2 in WEEE Reprocessing. Mater. Trans. B, vol. 52B (2), 871-882.
- Piskunen P., Avarmaa K., O’Brien H., Klemettinen L., Johto H. & Taskinen P. (2018): Precious Metals Distributions in Direct Nickel Matte Smelting with Low-copper Mattes. Mater. Trans. B., vol. 49 (1) 98-112.
- In your aims paragraph, on rows 49-52, you refer to distributions of PMs and PGMs between matte and slag which commonly are understood as a relation between chemically dissolved metals in matte and slag. Here, you discuss about total concentrations of the valuable metals in slag and matte, i.e. including mechanical matte dispersion in the slag together with the chemically dissolved species, which is very misleading. Thus, the paragraph must contain an explanation of the specific distribution information discussed in this manuscript.
- The above comment is valid with all the other elements and their distributions between slag and matte, i.e. with copper, nickel, and sulphur. Therefore, their chemical dissolution, depending on the sulphur and oxygen partial pressures, and mechanically dispersed fraction must be treated as separate element populations.
- The Materials and Methods Ch should include more details of the used materials, furnace and crucible construction, and execution of the experiments which obviously in case of sulphides were carried out in air.
- On rows 59-60, you very shortly describe the preparation technique used for synthetising matte. More details are needed for understanding the procedures used.
- On row 67, you specify the ‘accuracy’ of the ICP analyser without specifying whether that is the detection limit (of some elements) or an uncertainty of the analyses. Please, rephrase and update.
- In Results and Discussion Ch, you claim (rows 84-85) that most gold was concentrated in matte but do not disclose the analysed concentrations in slag. Perhaps, you should also point out that no kind of separation of mechanically dispersed matte from the slag was done prior to the chemical analyses of the slag and correspondingly the matte.
- The sentence on rows 98-100 is very unclear and must be rephrased. It is not evident either, how solubility of gold in matte at low temperatures (which has not treated in the Introduction as background information) links with chemical dissolution and entrainment of gold in slag at high temperatures.
- On rows 107-110, you introduce polished sections from the slag in pyrite smelting. How the CaS has been generated in a material with rather small CaO concentration (1.1 %) in oxidising conditions (air)? You also find platinum in that slag even if the raw material has no platinum!
- The captions of figures 1-3 should be more specific in illustrating the samples/raw materials of the mattes and slags described in the micrographs.
- On rows 128-129, you refer to copper-nickel matte smelting and its product in Fig. 2. This is confusing as the text above linked the very micrograph with the pyrite smelting. Please, refine and rephrase the text.
- In Fig. 3, you show metallic precipitates in the copper-nickel matte smelted in air. How they have been generated in the smelting?
- In table 3, please reorganise the caption and simplify the variable names so that the table is easy to read.
- On row 245 you use the phrase ‘ore-thermal furnace’ without explaining its meaning. Please, upgrade the text.
- On row 244, you give an estimate of precious metals distributions in the slag; it would be useful to discuss also the forms of platinum and gold present, e.g. their speciation.
- Therefore, captions of the Figures 5 and 6 must be updated as the abscissa axis only is an indication of dispersed matte in the slag, and it has no connection to chemical solubility of sulphur in the silicate melt/slag.
- On rows 266-267, you claim that CaO lowers the sulphur solubility in the matte. How you define here ‘desulphurization’ and how did you observe the process in a system reacting in air?
- On rows 269-277, you describe equilibrium calculations done using HSC Chemistry in reducing conditions. How they resembled your experimental conditions in air?
- On row 280 you describe melting point measurements of the ‘sulphide ore’. In Experimental Ch, there is no description on how melting point measurements were carried out!
- Table 4 shows the slag and matte assays in melting experiments without and with limestone. A similar table is needed about the CaF2
- On rows 300-301 you claim that gold and platinum exist in sulphide ores as metals. How did you confirm that in this study and with the concentrates used?

Author Response
Dear Reviewer,
Thank you for your attention to our manuscript ID metals-1405841 “Transfer of gold, platinum and non-ferrous metals from matte to slag during pyrometallurgical processing of raw materials”. We took into account your comments and made the appropriate changes and additions to the text (highlighted in red).
- The title of your manuscript is misleading the readers to kinetics and mass transfer between slag and matte; I suggest that you rewrite it in a more focused/shorter form.
We decided to change the title to more specific “Transfer of gold, platinum and non-ferrous metals from matte to slag by flotation”.
- In the Abstract (rows 13-14), you claim that bubbles appear in smelting because of decomposition of sulphides. This is only one form of gas possibly present in the sulphide smelting environments. Please, re-phrase.
The sentence “The main reason for their transition to slag is the flotation of sulfide droplets by gas bubbles that appear due to the decomposition of sulfides” has been re-placed by “The main reason for their transition to slag is the flotation of sulfide droplets by gas bubbles, a significant proportion of which occurs during the decomposition of sulfides”.
- The Introduction Ch deals a little with the (chemical) solubilities of precious and platinum group metals in smelting slags. Such background information and a clear definition of slag losses as mechanical and chemical would be useful when evaluating the claims and observations made in this study. Such data on chemical dissolution are available in several recent publications, e.g. Chen M., Avarmaa K., Klemettinen L., O’Brien H., Shi J., Taskinen P., Lindberg D. & Jokilaakso A. (2021): Precious Metal Distributions Between Copper Matte and Slag at High PSO2 in WEEE Reprocessing. Mater. Trans. B, vol. 52B (2), 871-882. Piskunen P., Avarmaa K., O’Brien H., Klemettinen L., Johto H. & Taskinen P. (2018): Precious Metals Distributions in Direct Nickel Matte Smelting with Low-copper Mattes. Mater. Trans. B., vol. 49 (1) 98-112.
Note taken into account. The references to relevant articles are inserted in the text.
- In your aims paragraph, on rows 49-52, you refer to distributions of PMs and PGMs between matte and slag which commonly are understood as a relation between chemically dissolved metals in matte and slag. Here, you discuss about total concentrations of the valuable metals in slag and matte, i.e. including mechanical matte dispersion in the slag together with the chemically dissolved species, which is very misleading. Thus, the paragraph must contain an explanation of the specific distribution information discussed in this manuscript. The above comment is valid with all the other elements and their distributions between slag and matte, i.e. with copper, nickel, and sulphur. Therefore, their chemical dissolution, depending on the sulphur and oxygen partial pressures, and mechanically dispersed fraction must be treated as separate element populations.
Note taken into account. The purpose of the work is rephrased as follows: “The purpose of the present work: to search for ways to increase the degree of extraction of platinum, gold and non-ferrous metals, experimentally study the effect of fluxes - calcite and fluorite on their distribution between matte and slag, estimated by the total metal content in slag and matte, based on theoretical concepts of droplet flotation”.
- The Materials and Methods Ch should include more details of the used materials, furnace and crucible construction, and execution of the experiments which obviously in case of sulphides were carried out in air.
Note taken into account. The Materials and Methods Ch has been expanded.
- On rows 59-60, you very shortly describe the preparation technique used for synthetising matte. More details are needed for understanding the procedures used.
Added the following sentences: “Copper-nickel matte was obtained as follows. Copper-nickel sulfide ore was melted in corundum crucibles (volume 100-150 ml) in a resistance furnace with a graphite heater at a temperature of 1300℃ and holding for 30 minutes. Heating to the specified temperature lasted 1.5-2 hours, cooling took place together with the furnace for 2-3 hours. After that, the matte was separated from the slag, the matte was crushed to a particle size of less than 0.071 mm, and re-melting was performed (to completely remove the slag particles). The re-melting conditions are like the first”.
- On row 67, you specify the ‘accuracy’ of the ICP analyser without specifying whether that is the detection limit (of some elements) or an uncertainty of the analyses. Please, rephrase and update.
Added the following sentence: “The detection limit for chemical elements is 0.1-0.3 ppm (g/t)”.
- In Results and Discussion Ch, you claim (rows 84-85) that most gold was concentrated in matte but do not disclose the analysed concentrations in slag. Perhaps, you should also point out that no kind of separation of mechanically dispersed matte from the slag was done prior to the chemical analyses of the slag and correspondingly the matte.
Note taken into account. The necessary explanations have been included in the text. The concentration of total gold in the slag reaches a maximum of 200 g/t.
- The sentence on rows 98-100 is very unclear and must be rephrased. It is not evident either, how solubility of gold in matte at low temperatures (which has not treated in the Introduction as background information) links with chemical dissolution and entrainment of gold in slag at high temperatures.
Earlier, we found that the amount of metallic impurities in dispersed particles of ore gold (in particular, in gold-bearing copper-pyrite ore, Table 1) increases with decreasing their size (with a particle size less than 3 μm). In our opinion, the size effect is associated with the amorphization of particles smaller than 3 μm, a sharp change in the structure and properties in comparison with the bulk material and, as a consequence, an increase in the solubility of metals in gold due to an increase in the Gibbs energy. Thus, the increase in the amount of impurities is associated with the size factor, which will take place in both ores and matte.
An explanation has been inserted into the text.
- On rows 107-110, you introduce polished sections from the slag in pyrite smelting. How the CaS has been generated in a material with rather small CaO concentration (1.1 %) in oxidising conditions (air)? You also find platinum in that slag even if the raw material has no platinum!
The CaS label has been removed.
- The captions of figures 1-3 should be more specific in illustrating the samples/raw materials of the mattes and slags described in the micrographs.
Note taken into account. The captions of figures 1-3 have been expanded and concretized.
- On rows 128-129, you refer to copper-nickel matte smelting and its product in Fig. 2. This is confusing as the text above linked the very micrograph with the pyrite smelting. Please, refine and rephrase the text.
Note taken into account. Figure 2 is divided into 2 separate figures and the corresponding explanations are included in the text.
- In Fig. 3, you show metallic precipitates in the copper-nickel matte smelted in air. How they have been generated in the smelting?
We did not study the mechanism of formation of platinum-iron alloys, like all other phases. The figure was obtained after cooling the melt.
- In table 3, please reorganise the caption and simplify the variable names so that the table is easy to read.
Note taken into account. The edits are included in table 3.
- On row 245 you use the phrase ‘ore-thermal furnace’ without explaining its meaning. Please, upgrade the text.
Note taken into account. The sentence is rephrased as follows: “In industrial slags in ore-thermal furnaces when melting on copper-nickel matte, the content of platinum ranges from 0.3 to 1 g/t, gold from 0.1 to 0.4 g/t”.
- On row 244, you give an estimate of precious metals distributions in the slag; it would be useful to discuss also the forms of platinum and gold present, e.g. their speciation.
The corresponding reference is included in the text. The following sentence has been inserted: “In the slags formed after the processing of copper-nickel sulfide ores, platinum is in the form of alloys with iron and nickel”.
- Therefore, captions of the Figures 5 and 6 must be updated as the abscissa axis only is an indication of dispersed matte in the slag, and it has no connection to chemical solubility of sulphur in the silicate melt/slag.
The analysis was performed for total sulfur (including both chemically dissolved sulfur and sulfur in matte droplets). The necessary explanation is included in the text and in Figures 5 and 6.
- On rows 266-267, you claim that CaO lowers the sulphur solubility in the matte. How you define here ‘desulphurization’ and how did you observe the process in a system reacting in air?
That the presence of CaO, which reacts with compounds containing sulfur, and therefore reduces its content, is confirmed by reference. We did not deal with the details of this process in this work. Our early research is presented in the article: “Amdur, A.M.; Blagin, D.V., Pavlov, V.V.; Munkhtuul, L. Removal of sulfur in the reduction of iron-ore concentrates by coal. Coke and Chemistry 2013, 3, 2-6. DOI: 10.3103/S1068364X13030034”.
- On rows 269-277, you describe equilibrium calculations done using HSC Chemistry in reducing conditions. How they resembled your experimental conditions in air?
Since the heating and melting of the material took place in a furnace with a graphite heater, carbon-containing gases were released, which could interact with the test material. Therefore, the modeling was carried out in the presence of carbon in the system. The forms of occurrence of gold, platinum and non-ferrous metals are consistent with experimental results.
- On row 280 you describe melting point measurements of the ‘sulphide ore’. In Experimental Ch, there is no description on how melting point measurements were carried out!
A W-Re thermocouple in a corundum sheath was in contact with the charge. At the indicated temperature, a break was observed on the temperature - heating curve, which we interpreted as melting of the charge. This is confirmed by visual observations. The corresponding explanation is inserted into the text.
- Table 4 shows the slag and matte assays in melting experiments without and with limestone. A similar table is needed about the CaF2
The table (CaF2) has been added to the text.
- On rows 300-301 you claim that gold and platinum exist in sulphide ores as metals. How did you confirm that in this study and with the concentrates used?
A typo is allowed. It has been corrected to the following text: “Gold and platinum in the studied products of sulfide ores smelting are in the form of metallic phases containing metallic impurities”.
Best regards,
A team of authors.

Reviewer 2 Report
In the present study, the authors introduced two fluxes (CaCO3 and CaF2) into the initial charge during melting of sulfide materials onto matte to reduce the content of sulfide drops (including non-ferrous matals, gold and platinum) in the slag. This article is well-organized with innovation, but the following issues need to be addressed.
- The resolution ratio of some figures are too low to see the details (Figs. 4, 5, 6). The last line of all tables are missing.
- The introduction can be extented as there are only several references cited. The authors can put more content in here to improve their research impact.
- Please add appropriate numbers of sub-sections to Section 2 and 3, as there are too many contents in one section.
- Please give the reason that why CaCO3 and CaF2 as fluxes were chosen in this study?
- If possible, I suggest the authors to add one figure about the role of these two fluxes considering Fig. 4 only provide gas bubble through the matte-slag interface.
Author Response
Dear Reviewer,
Thank you for your attention to our manuscript ID metals-1405841 «Transfer of gold, platinum and non-ferrous metals from matte to slag during pyrometallurgical processing of raw materials». We took into account your comments and made the appropriate changes and additions to the text.
- The resolution ratio of some figures are too low to see the details (Figs. 4, 5, 6). The last line of all tables are missing.
Note taken into account. The quality of the figures 4-6 has been improved.
- The introduction can be extented as there are only several references cited. The authors can put more content in here to improve their research impact.
Note taken into account. The number of references has been increased.
- Please add appropriate numbers of sub-sections to Section 2 and 3, as there are too many contents in one section.
Note taken into account. The Results and Discussions section is divided into several sub-sections.
- Please give the reason that why CaCO3 and CaF2 as fluxes were chosen in this study?
CaCO3 is the source of many bubbles during its dissociation (CO2). These bubbles in the slag can float and coagulate the matte microdroplets. CaCO3 is a common and readily available flux (limestone). CaF2 can significantly lower the viscosity of the melt (by an order of magnitude, which happened in our case).
- If possible, I suggest the authors to add one figure about the role of these two fluxes considering Fig. 4 only provide gas bubble through the matte-slag interface.
The addition of the listed fluxes will lead to a change in interfacial tensions, a decrease in viscosity and, accordingly, an increase in the sedimentation rate of matte drops. In this case, the circuit itself, shown in figure 4, will not change.
Best regards,
A team of authors.

Round 2
Reviewer 1 Report
. Its lay-out needs still various modifications in order to avoid extension of tables and figure captions on two pages. There is also some tautology in captions of Figs. 1-3 on the SEM used but the Experimental Ch, on the other hand, fails in reporting the operational conditions of it.

Author Response
Dear Reviewer,
Thank you for your attention to our manuscript ID metals-1405841 “Transfer of gold, platinum and non-ferrous metals from matte to slag by flotation”. We took into account your comment and made the appropriate changes to the text.
- Its lay-out needs still various modifications in order to avoid extension of tables and figure captions on two pages. There is also some tautology in captions of Figs. 1-3 on the SEM used but the Experimental Ch, on the other hand, fails in reporting the operational conditions of it.
Some of the lines in table 2 have been removed. The captions in figures 1-4 have been corrected. Added the following text to the Materials and Methods chapter: “From the other part, polished sections (3×2 cm in size) were made, which were then examined under an Axio Image optical microscope and a Tescan Vega 3 scanning electron microscope equipped with an Oxford Instruments X-act energy dispersive analyser. SEM photography was performed at high vacuum, voltage 20 kV and backscattered electrons mode (contrast mode by average atomic number). The energy dispersive analyzer allows the determination of chemical elements with an accuracy and detection limit of 0.1 wt%.
Best regards,
A team of authors.